# *Clostridium difficile* Infection and Colorectal Surgery: Is There Any Risk?

**DOI:** 10.3390/medicina55100683

**Published:** 2019-10-10

**Authors:** Valentin Calu, Elena-Adelina Toma, Octavian Enciu, Adrian Miron

**Affiliations:** 1Elias University Emergency Hospital, 011461 Bucharest, Romania; drcalu@yahoo.com (V.C.); esoctavian@gmail.com (O.E.); dramiron@yahoo.com (A.M.); 2Carol Davila University of Medicine and Pharmacy, 020021 Bucharest, Romania

**Keywords:** *Clostridium difficile* infection, colorectal surgery, anastomotic leakage

## Abstract

*Background and objectives: Clostridium difficile* infection (CDI) is an important healthcare-associated infection, with important consequences both from a medical and financial point of view, but its correlation with anastomotic leaks after colorectal surgeries is scarcely reported in the literature. *Materials and Methods:* We conducted a retrospective study looking for patients who underwent open or laparoscopic surgery for colorectal cancers between January 2012 and December 2017, excluding emergency surgeries for complicated colorectal tumors. We also examined patient history for risk factors for CDI such as age, sex, comorbidities, and clinical findings at admission or during hospital stay as well as tumor characteristics. *Results:* A total of 360 patients were included in the study, out of which 320 underwent surgeries that included anastomoses. There were 19 cases of anastomotic leaks, out of which 13 patients were diagnosed with CDI, with a statistic significance for association between CDI and anastomotic leakage (*p* < 0.0001). Most patients who developed both CDI and anastomotic leaks had left-sided resections or a type of rectal resection, while none of the patients with right-sided resections had this association, but with no statistical significance possibly due to the limited number of cases. *Conclusions:* CDI is a relevant risk factor and should be taken into consideration when trying to prevent anastomotic leaks in patients undergoing gastrointestinal surgery for colon or rectal cancer. Thorough assessment of risk factors at admission should be mandatory in order to adequately prepare the patient and plan an optimal course of treatment. Further studies are needed to confirm our findings and a multidisciplinary approach, with a team which should always include the surgeon, is mandatory when it comes to CDI prevention.

## 1. Introduction

*Clostridium* spp. infection is a common problem nowadays in all surgical and gastroenterology departments. The consequences are sometimes severe from a medical, financial, and legal point of view. *Clostridium* spp. have been identified more frequently in patients with inflammatory bowel disease and have been reported less frequently to increase morbidity and mortality after endoscopic procedures with no visible perforation but with postprocedural bacteremia [1,2,3].

*Clostridium difficile*, first described by Hall and O’Toole in 1935, is an anaerobic Gram-positive bacillus that can form heat-resistant spores which are very persistent especially in environments such as hospitals and other healthcare facilities [4].

Scarce published data document the relationship between *C. difficile* infection (CDI) and anastomotic leakage in colorectal surgery. Nevertheless, it should not be disregarded when searching for suspects in the unfavorable postoperative course of surgical patients, especially those with important comorbidities and known risk factors for anastomotic leaks. While age, antibiotic (especially combined antibiotics) and proton pump inhibitor use in the 10 days prior to surgery, previous hospitalization, and decreased immunity are firmly established as favorable factors in developing postoperative *C. difficile* colitis, the role of bowel preparation remains controversial [5,6]. Colorectal surgery, in particular, is seldom cited as an independent risk factor in up to 21% of patients who had undergone colectomies and 29% of patients who underwent a surgical procedure for bowel obstruction, but the complexity of the surgery itself, as represented by operative time and blood loss, did not correlate with increased odds of CDI [7].

## 2. Material and Methods

We have retrospectively studied, using operative logs, the patients that underwent open or laparoscopic surgery for colorectal cancer between January 2012 and December 2017, excluding emergency interventions for complicated colorectal tumors. The study was approved by the institutional ethics committee and review board of the Elias University Emergency Hospital (approval number 9299, approved on 28 May 2019) and was conducted in accordance with the ethical principles stated in the Declaration of Helsinki and informed consent was obtained from all the patients included in the study. The authors declare that no experiments were performed on humans or animals for this study, in accordance with The Code of Ethics of the World Medical Association (Declaration of Helsinki). The authors declare that they have followed the protocols of their work center on the publication of patient data. The authors declare that no identifiable patient data appear in this article. Medical records were then reviewed to find patients who developed anastomotic leaks after surgery as well as patients diagnosed with *Clostridium difficile* infection during their stay in the same hospital (both toxins A and B, using an enzyme immunoassay). Patients who tested positive for *Clostridium difficile* at admission or prior to surgery and those who underwent interventions that did not involve the colon or rectum were excluded. Patient history was audited for risk factors such as age, sex, comorbidities, and clinical findings at admission (anemia, hypoalbuminemia, and electrolyte imbalance) or during hospital stay (postoperative abdominal pain, diarrhea, fever, and leukocytosis) as well as tumor characteristics and intraoperative findings. We also compared the medication they received prior to and after the surgery as well as their clinical course.

Patients undergoing elective colorectal surgery had received preoperative bowel preparation as well as prophylactic therapy with oral antibiotics in the 18 h prior to surgery (rifampicin and metronidazole single dose) and intravenous ceftriaxone and metronidazole or ertapenem single dose 30 min before surgery.

For statistical analysis, the SPSS version 20.0 (SPSS Inc., Chicago, IL, USA) was used. Cases with *P*-values lower than 0.05 were considered to be statistically significant.

## 3. Results

A total of 360 patients were included in the study, out of which 242 had colon tumors and 118 presented with rectal cancer. Of the 242 patients, 70 underwent right hemicolectomy, 25 extended right hemicolectomy, 11 transverse colon resection, 13 splenic flexure resection, 53 left hemicolectomy, 41 sigmoid resection, 8 subtotal colectomy, 6 total colectomy, and 10 Hartmann’s procedure. Of the patients diagnosed with rectal cancer, 58 suffered low anterior resection, 6 ultra-low anterior resection with coloanal anastomosis, 19 Hartmann’s procedure, and 30 abdominoperineal resection. Ten patients underwent different types of colic resections with the formation of loop stoma. Thirty-seven patients underwent laparoscopic resections. Of the 360 cases, 320 had anastomoses: 293 per primam, 23 went on to have surgery for restoration of continuity, and four had delayed coloanal anastomosis. Fifty-one of the anastomoses were done using mechanical sutures and the rest, 269, were hand-sewn. There were 19 cases of anastomotic leaks, meaning 6% out of those 320 who had anastomoses.

When assessing risk factors for developing CDI (Table 1), we noticed that 53.8% of the patients were over 60 years of age, most of them had stayed in the hospital for longer than 10 days (mean value of 11.6), and over half of them were anemic and hypoalbuminemic at the time of admission, but only 6.8% had recently taken proton pump inhibitors (less than seven days prior to surgery). Most importantly, almost 40% had received perioperative antibiotic treatment, with 21 patients being treated with multiple antibiotics for various outpatient conditions (penicillins, quinolones, and second-generation cephalosporines).

Twenty-eight patients (7.77% of all patients, 8.75% of patients with anastomoses) tested positive for *Clostridium difficile* infection (CDI) during the postoperative hospital stay. The mean age of the 28 patients was 62 (range 47–82 years) and the male to female ratio was 1:1. The mean hospital stay was 19.3 days (range 14–32 days). Ten patients (35.7%) were anemic at the time of admission, six (21.4%) had hypoalbuminemia, and five (17.8%) had altered electrolyte balance. Recent proton pump inhibitor intake (less than seven days prior to surgery) was documented in two patients (7.1%). Sixteen patients (57.1%) received preoperative antibiotics (second-generation cephalosporines) or postoperative iv antibiotic treatment (cephalosporins or carbapenems), but before the detection of the *Clostridium difficile* toxin. Thirteen patients received left hemicolectomy, five patients underwent right hemicolectomy, three patients needed sigmoid resections, and seven underwent low or ultra-low anterior rectal resections.

Mean onset-time of CDI was 6.5 days after surgery (range 1–24 days) and for three patients (10.7%) recurrent CDI was documented (previously diagnosed and treated for CDI less than 30 days prior to surgery). Diarrhea was present in all 28 patients (100%), 11 of them had leukocytosis (39%), with fever being observed in 8 cases (28.5%) and abdominal pain was reported in 5 patients (17.8%). Antibiotic therapy that had incited the CDI was interrupted in all patients to prevent CDI recurrence. Twenty-six of the patients (92.8%) received a 10-day combined treatment with vancomycin (125 mg orally 4 times per day) and metronidazole (500 mg orally 3 times per day). Considering previous treatment with vancomycin, 2 patients (7.2%) were given tigecycline (an initial dose of 100 mg followed by 50 mg twice daily) and metronidazole. None of the cases presented as fulminant CDI. The three patients diagnosed with recurrent CDI after initial antibiotherapy with vancomycin and metronidazole, received a standard 10-day course of vancomycin. There were no reinterventions required for CDI-related toxic megacolon. Clinical findings and treatment course are summarized in Table 2.

Thirteen patients diagnosed with CDI had undergone surgery that involved a type of anastomosis and developed subsequent anastomotic leakage, with a statistic significance for association between CDI and anastomotic leak (*p* < 0.0001) (Table 3). None of the patients in the right hemicolectomy group that developed CDI had anastomotic leaks. Eight had undergone low or ultra-low anterior resection (with stapled anastomoses), two were in the sigmoid resection group (hand-sewn anastomoses), and three had undergone left hemicolectomies (two hand-sewn and one stapled anastomosis). Although there was no need for reintervention for toxic megacolon caused by CDI, out of the 13 patients who developed anastomotic leaks and tested positive for CDI, only two who underwent low anterior resections were managed conservatively with aggressive antibiotherapy and consequently had a favorable outcome. The remaining eleven patients underwent re-laparotomy with subsequent defunctioning of the anastomosis or had the anastomosis taken down with permanent colostomy formation, as well as postoperative admission in the ICU.

Five of the patients died of causes related to the initial disease or complications that arose from it, but worsened by the CDI (17.8% of all patients with CDI, 38.46% of patients who had anastomotic leak and CDI). One of them was a 47-year-old man who underwent an ultra-low anterior resection, but had no risk factors for developing CDI at admission. Other two patients with ultra-low anterior resections, one patient with sigmoidectomy, and one patient with left hemicolectomy that did not survive after reintervention were over 60, hypoalbuminemic, with electrolyte imbalance at admission, and had received combined oral antibiotics prior to surgery.

## 4. Discussion

*Clostridium difficile* infection is an increasingly common nosocomial occurrence, with a reported general incidence of up to 7.8%, while its incidence in patients undergoing colorectal surgery is almost three times higher [4,8,9,10]. Although it was initially considered a non-threatening bacterium, as it was discovered in the microbial flora of healthy newborns, *Clostridium difficile* has proven itself to be a very dangerous pathogen especially in the era of increased use of broad-spectrum antibiotics, most often clindamycin, penicillins, and cephalosporins, which alter normal colonic flora and cause pseudomembranous colitis or mild to severe forms of diarrhea without colitis [10,11].

The bacillus produces two cytotoxins, type A and type B, which trigger a proinflammatory response in white blood cells, leading to the particular aspect of the colonic mucosa [4,12]. In our study, all 28 patients had the type A toxin diagnosed by an immunoassay. It is important to mention that not all strains of *Clostridium difficile* are equally virulent, with community-acquired strains being far less likely to cause severe manifestations of the disease and also showing less resistance to treatment, while recently described strains such as ribotype 027 are associated with higher mortality rates especially in hospitalized patients [6,12]. We were unable to identify the specific ribotype of *C. difficile* toxins due to technical limitations of in-hospital laboratories.

While up to 50% of hospitalized patients may be colonized with the bacillus (13% of patients admitted for up to two weeks and 50% for those admitted for up to four weeks), half of them remain asymptomatic [13,14]. Diarrhea (>3 bowel movements over 24 h) is regarded as the trademark of *Clostridium difficile* infection, although it might be absent or might appear late in the development of the disease due to concomitant ileus, and is often associated with moderate to intense abdominal pain, nausea, anorexia, and if left untreated can lead to electrolyte imbalance and dehydration [14]. All the patients diagnosed with CDI in our study had diarrhea, rather than presenting with ileus. Predictive factors for severe infection include fever (>38.5 °C), leukocytosis (>15,000/mm^3^), elevated serum creatinine levels, and hypoalbuminemia (<2.5 mg/dL) [11]. The infection can escalate to complications such as toxic megacolon with colonic perforation, peritonitis, septic shock, and subsequent organ failure. Typical lesions are visible during sigmoidoscopy or colonoscopy—adherent yellow plaques of variable sizes mainly involving the rectum and sigmoid. [14,15]. Symptomatic patients generally have at least one associated risk factor: age over 60, compromised immunity, renal failure, prolonged hospital stay, antibiotic treatment, or gastrointestinal surgery, which was also confirmed in our group [16]. While 12 of the 28 patients diagnosed with CDI had received preoperative antibiotics with second-generation cephalosporins, we could only assess the scope of this treatment in five of them (urinary tract infections, otitis media, and sinusitis), the others having been prescribed a full course by their General Practitioner for uncertain afflictions.

Recent published data indicate that treatment with proton pump inhibitors or prolonged perioperative antibiotherapy (previously regarded as safe or non-related) might be substantial contributors to outbreaks of virulent strains [7,10]. There are still debates regarding mechanical bowel preparation in association with prophylactic intravenous or oral antibiotherapy as a risk factor for developing *Clostridium difficile* infection in patients undergoing colorectal surgery [17,18]. All patients included in our study had preoperative mechanical bowel preparation, but no antibiotic bowel preparation.

There is a rising concern that *Clostridium difficile* infection is an important contributor to anastomotic leaks following colorectal surgery, especially after anterior resections for rectal cancer [13]. In our study, most patients who developed both CDI and anastomotic leaks had a type of rectal resection, but with no statistical significance possibly due to the limited number of cases. The rate of anastomotic leakage after such procedures is variably reported, between 2.3% and 19.2% of patients, with even more for hand-sewn versus stapled anastomoses [19]. Risk factors for anastomotic leakage are related to patient (ASA score, age >75, male sex, obesity, hypoalbuminemia, anemia, and comorbidities), disease-related (advanced-stage cancer and lowly situated tumors), and surgical approach (emergency surgery, prolonged operative time, low level of anastomosis, drainage, and surgeon’s inexperience) [16,20].

Although our study has the limitations of a retrospective analysis, such as incomplete data acquisition, we were able to analyze data that produced statistically significant correlations between CDI and anastomotic leakage. Limitations also include incomplete medical history and uncertain previous treatment with empiric antibiotherapy prescribed in outpatient clinics or GP practices as well as self-prescribed over-the-counter proton pump inhibitors intake. The role of bowel preparation (combined mechanical/oral antibiotic bowel preparation) versus no preparation in the subsequent development of CDI was beyond the scope of our study.

In order to audit the impact of CDI in colorectal surgery, a multicentric registry would be beneficial. This would aid with large case volume analysis to shed light on the role of this hospital-acquired or health-care associated infection in the postoperative course of colorectal surgical patients.

## 5. Conclusions

CDI is a relevant risk factor and should be taken into consideration when trying to prevent anastomotic leaks in patients undergoing surgery for colon or rectal cancer. Careful and thorough assessment of risk factors for CDI at admission is also advisable in order to personalize preoperative care and surgical management to better suit the status of the patient. Moreover, it is not only a nuisance for the surgeon and the patient, but also a burden in regard to both costs and prevalence of multi-resistant bacteria associated with medical procedures. Further studies are needed to confirm our findings and a multidisciplinary approach with a team which should include the surgeon as well is mandatory when it comes to CDI prevention.

## Figures and Tables

**Table 1 medicina-55-00683-t001:** Risk factors for *Clostridium difficile* infection (CDI; *n* = 320).

	No. of Patients (%)
Age	
<60	148 (46.2)
>60	172 (53.8)
Sex	
Male	181 (56.5)
Female	139 (43.5)
Hospital stay (days)	11.6 (5–41) *
Laboratory tests	
Anemia	236 (73.7)
Hypoalbuminemia	165 (51.5)
Electrolyte imbalance	121 (37.8)
Recent IPP intake	22 (6.8)
Antibiotic treatment	126 (39.3)
Preoperative oral antibiotics	107 (33.4)
Postoperative single-dose	12 (3.7)
Postoperative prophylactic treatment	7 (2.1)
Administration of multiple antibiotics	21 (6.5)

* Value representing mean (range).

**Table 2 medicina-55-00683-t002:** Clinical findings and treatment for CDI (*n* = 28).

	No. of Patients (%)
Age	
<60	10 (35.7)
>60	18 (64.3)
Sex	
Male	14 (50)
Female	14 (50)
Hospital stay (days)	19.3 (14–32) *
Laboratory tests	
Anemia	10 (35.7)
Hypoalbuminemia	6 (21.4)
Electrolyte imbalance	5 (17.8)
Recent IPP intake	2 (7.1)
Antibiotic treatment	16 (57.1)
Preoperative oral antibiotics	12 (42.8)
Postoperative single-dose	1 (3.5)
Postoperative prophylactic treatment	3 (10.7)
Type of resection	
Right hemicolectomy	5 (17.8)
Left hemicolectomy	13 (46.4)
Sigmoid resection	3 (10.7)
Low/Ultra-low anterior resection	7 (25)
Onset time of *C. difficile* colitis (day)	6.5 (1–24) *
Recurrent CDI	3 (10.7)
Diarrhea	28 (100)
Leukocytosis	11 (39)
Fever	8 (28.5)
Abdominal pain	5 (17.8)
Treatment	
Vancomycin + Metronidazole	26 (92.8)
Tigecycline + Metronidazole	2 (7.2)
Vancomycin (recurrent CDI)	2 (7.2)

* Value representing mean (range).

**Table 3 medicina-55-00683-t003:** Comparison of anastomotic leakage in patients with and without *Clostridium difficile* colitis.

	*C. difficile* Positive (No. of Patients)	*C. difficile* Negative (No. of Patients)	Incidence (%)
Anastomotic leak	13	6	68.4 *
No leak	15	286	4.98

* *p* < 0.0001; RR = 13.7 (7.68 < RR < 24.52).

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
