# Peer review of "Clostridium difficile Infection and Colorectal Surgery: Is There Any Risk?"

_medicina, 2019, doi:10.3390/medicina55100683_

Round 1

Reviewer 1 Report

The authors correlated anastomotic  leakages to  Clostridioides difficile infection in patients who underwent colorectal surgery. Although C.difficile is considered as a major cause of Hospital acquired infection, there are several studies which have already discussed the role of colorectal surgeries in initiating this infection. Analysis of  more number of cases with anostomotic leakage, hemicolectomy and resection surgeries would shed more light on C.difficile associated infection in these subgroups of patients.

Author Response

Thank you for your review. 

We extended data presented in the introduction and presented the limitations to our study in a more eloquent manner, meanwhile adding the suggestion of creating a registry that would help analyse more cases in the future in order to assess the validity of the correlation we found. 

We reviewed the English used throughout the paper as per instructed.

Reviewer 2 Report

Thank you for this interesting paper.

I only have a few more suggestions:

INTRODUCTION

Although the study focuses on Clostridium difficile, an overview of Clostridium spp infections should be given in the introduction. In fact Clostridium spp. infections do not only complicate surgery, but can also result from other procedures (e.g. endoscopy), in a very peculiar way. With regard to that, the authors should include in the overview on Clostridium spp, the following papers:

Gioia S, Lancia M, Mencacci A, Bacci M, Suadoni F. Fatal Clostridium perfringens Septicemia After Colonoscopic Polypectomy, Without Bowel Perforation. J Forensic Sci. 2016 Nov;61(6):1689-1692. Shaw E, Reyes R, Bonet A, Garcia-Huete L, Pasqualetto A, Tubau F, et al. Fatal retroperitoneal gas gangrene complicating colonoscopic polypectomy without bowel perforation in a healthy adult. Endoscopy 2014;46:e91–2. Boenicke L, Maier M, Merger M, Bauer M, Buchberger C, Schmidt C, et al. Retroperitoneal gas gangrene after colonoscopic polypectomy with- out bowel perforation in an otherwise healthy individual: report of a case. Langenbecks Arch Surg 2006;391(2):157–60.

LIMITATIONS

Limitations should be better explained in a separate paragraph after results.

DISCUSSION

The “discussion” section has to contain statements of strength, limitations, novelty, reason for publishing and practical implications.

The results should be more discussed in the “discussion” section.

Author Response

Thank you for your review. 

We included additional information in the introduction, citing the three recommended articles and explaining the framing and reasoning for the paper. 

We clearly mentioned the limitations we found during the study, in the results section and the discussions section, as highlighted. 

We explained the results we came to in a more detailed manner and made correlations with clinical implications and the need for further studies in the future, as well as suggestions for improvement of the current status. 

Round 2

Reviewer 1 Report

Thank You for providing the revised manuscript for review.

The initiative to launch a multidisciplinary study on colorectal surgery and infection is laudable. This study should be a stepping stone towards a more elaborate study on Clostridium infections in colorectal surgery.

English language & style check, spell check is essential.

Author Response

Thank you for your appreciation and suggestions. 

We revised the writing style and did a thorough grammar and spelling check and made the necessary changes.